# Naphtho-Gamma-Pyrones Produced by *Aspergillus tubingensis* G131: New Source of Natural Nontoxic Antioxidants

**DOI:** 10.3390/biom10010029

**Published:** 2019-12-24

**Authors:** Quentin Carboué, Marc Maresca, Gaëtan Herbette, Sevastianos Roussos, Rayhane Hamrouni, Isabelle Bombarda

**Affiliations:** 1Vinovalie, ZA les portes du Tarn, 81370 Saint-Sulpice-la-Pointe, France; 2Aix Marseille Univ, Avignon Université, CNRS, IRD, IMBE, 13397 Marseille, France; sevastianos.roussos@imbe.fr (S.R.); rayhan.hamrouni@gmail.com (R.H.); 3Aix Marseille Univ, CNRS, Centrale Marseille, iSm2, 13397 Marseille, France; 4Aix Marseille Univ, CNRS, Centrale Marseille, FSCM, Spectropole, 13397 Marseille, France; gaetan.herbette@univ-amu.fr

**Keywords:** naphtho-gamma-pyrones, radical scavenging activity, hydrogen peroxide, mediated cell death assay, solid state fermentation, *Aspergillus tubingensis*

## Abstract

Seven naphtho-gamma-pyrones (NγPs), including asperpyrone E, aurasperone A, dianhydroaurasperone C, fonsecin, fonsecinone A, fonsecin B, and ustilaginoidin A, were isolated from *Aspergillus tubingensis* G131, a non-toxigenic strain. The radical scavenging activity of these NγPs was evaluated using ABTS assay. The Trolox equivalent antioxidant capacity on the seven isolated NγPs ranged from 2.4 to 14.6 μmol L^−1^. The toxicity and ability of the NγPs to prevent H_2_O_2_-mediated cell death were evaluated using normal/not cancerous cells (CHO cells). This cell-based assay showed that NγPs: (1) Are not toxic or weakly toxic towards cells and (2) are able to protect cells from oxidant injuries with an IC_50_ on H_2_O_2_-mediated cell death ranging from 2.25 to 1800 μmol mL^−1^. Our data show that *A. tubingensis* G131 strain is able to produce various NγPs possessing strong antioxidant activities and low toxicities, making this strain a good candidate for antioxidant applications in food and cosmetic industries.

## 1. Introduction

Naphtho-gamma-pyrones (NγPs) are polyketide pigments that have been found to occur in higher plants belonging to the genera Cassia. They are also widely produced by fungi and, notably, *Aspergillus* spp. [1,2]. Their structures consist of a naphthalene and a γ-pyrone moiety, and there are monomeric and dimeric forms for which a diaryl bond links two NγPs together [3]. The NγPs’ dimerization is carried out by cytochrome P450 enzymes, which stereoselectivity depend on monomeric substrates [4]. This family exhibits a broad spectrum of biological activities, including antimicrobial, antiviral, insecticidal, anti-estrogenic, and antioxidant activity [5,6,7]. Antioxidants are compounds that can delay, inhibit, or prevent the oxidation of oxidizable matters by scavenging free radicals and diminishing oxidative stress. Such stress corresponds to an imbalanced state, where excessive quantities of reactive oxygen species (e.g., superoxide radical anion, hydrogen peroxide, and hydroxyl radical) are present at levels more than what is required for normal cell function and overwhelm endogenous antioxidant capacity and repair [8]. Nowadays, there is increasing interest in antioxidants, particularly in those intended to prevent the presumed deleterious effects of free radicals in the human body, and to prevent the deterioration of fats and other constituents of foodstuffs. In both cases, there is a preference for antioxidants from natural rather than from synthetic sources [9]. Few studies deal with the in vitro antioxidant activity (AOA) based on the trapping of the stable radical DPPH (2,2-diphenyl-1-picrylhydrazyl) and NγPs. Among the NγPs tested, some were reported to have or not antioxidant activity depending of the studies [10,11,12,13].

Among the published activities, only fonsecin shows a strong AOA compared to ascorbic acid, fonsecinone A having a moderate AOA and aurasperone A, a moderate or no significant AOA compared to pure references ascorbic acid or BHT. In the present study, we isolated various NγPs (including asperpyrone E, aurasperone A, dianhydroaurasperone C, fonsecin, fonsecinone A, fonsecin B, and ustilaginoidin A from *Aspergillus tubingensis* G131 and evaluated their AOA using the radical 2,2′-azino-bis(3-ethylbenzothiazoline-6-sulfonic acid radical cation (ABTS^●+^) assay, commonly used in the agrifood sector and H_2_O_2_-mediated cell death assay.

## 2. Materials and Methods

### 2.1. Nuclear Magnetic Resonance

Nuclear magnetic resonance (NMR) spectra were recorded with an Avance III Bruker NMR spectrometer operating at 600 MHz for ^1^H and 150 MHz for ^13^C. The chemical shifts are given in δ (ppm) and coupling constants (*J*) are reported in hertz (Hz). The one- (1D) and two-dimensional (2D) spectra were processed using Topspin. The NMR spectra were measured as solutions in chloroform-d or methanol-d following the solubility of the compounds in 5 mm outer diameter tubes. The structural identifications of the compounds were achieved by comparing their spectroscopic data (^1^H, ^13^C-NMR) with the values found in the literature.

### 2.2. Analytical HPLC Analysis

Dry samples were dissolved in ethanol and filtered. A C18 RP, Zorbax Eclipse XDB column 150 × 5 mm, 4.6 μm particle size (Agilent Technologies©, Santa Clara, CA, USA) fitted with a C18 guard column was used with a flow rate of 1 mL min^−1^ and an injection volume of 40 μL. Elution was performed using acidified water (H_2_O with 2% acetic acid) (solvent A) and acetonitrile (ACN) (solvent B) with a linear gradient over 55 min at a flow rate of 1 mL min^−1^, starting from 30 to 100% of solvent B during 45 min, followed by 100% solvent B for 5 min. The UV absorbance of the eluent was monitored at 280 nm.

### 2.3. Semi-Preparative HPLC Analysis

Separation of the compounds was performed using HPLC apparatus (Agilent Technologies©, USA). The column was a C18 (250 × 10 mm, 5 μm particle size) (Ascentis©, Eden Prairie, MN, USA) with a flow rate of 3 mL min^−1^ and an injection volume of 100 μL, following a linear gradient starting from H_2_O/ACN (70:30, *v*/*v*) to 100% ACN during 60 min. The UV absorbance of the eluent was monitored at 280 nm.

### 2.4. Fungal Material

*A. tubingensis* G131 was provided by École Nationale Supérieure Agronomique de Toulouse (France); this fungus is non-mycotoxigenic as the synthetic genes required for the production ochratoxin A and fumonisin are absent from its genome [14]. It was conserved at 4 °C in a 15 mL bottle on potato dextrose agar (PDA). The inoculum stock was prepared by propagating the fungus in 250 mL Erlenmeyer flasks containing PDA. The cultures were incubated at 25 °C for 5–10 days. The inoculum suspension was prepared by adding 0.01% (*v*/*v*) Tween 80 (Sigma-Aldrich, St. Louis, MO, USA) and scraping with a magnetic stirrer to recover the conidia. The quantity of conidia was counted using a Malassez cell prior to inoculation of the solid medium.

### 2.5. Solid State Fermentation

The SSF was performed in Raimbault columns on 50 g (dry material) of moist solid medium composed by 50% of vine shoots, 25% of wheat bran, and 25% of potato flakes. The medium was inoculated with 2.10^7^ conidia g^−1^ (dry material) at an initial humidity of 66% and incubated at 25 °C during 7 days.

### 2.6. Isolation of Compounds ***1**–**7***

One kilogram of moist fermented solid matrix was extracted using 2 L of ethanol at 25 °C during 2 h. The obtained extract was filtered with Whatman No. 1 filter paper and concentrated under reduced pressure using a rotary evaporator at 40 °C. The extract (8.32 g) was suspended in 250 mL of distilled water before being partitioned with hexane (84.7 mg), dichloromethane (334.9 mg), ethyl acetate (143.52 mg), and butanol (107.5 mg). The fractions that exhibited the most powerful antioxidant activity, thus the methylene chloride and the ethyl acetate extracts, were subsequently purified by semi-preparative HPLC. Twenty fractions were obtained (F1–F20), and subsequent semi-preparative HPLC on F6, F9, F16, F17, F19, and F20 allowed to isolate compounds **1**–**7**. Compounds **1** (16 mg) and **3**–**7** were isolated as yellowish powder (15, 9, 12, 7, 8, and 10 mg, respectively), whereas compound **2** (11 mg) was presented as red needles.

The spectroscopic data of the known NγPs were in accordance with literature.

*Fonsecin* (**1**). ^1^H-NMR (600 MHz, CD_3_OD) *δ*: 6.43 (1H, d, *J* = 2.1 Hz, H-9), 6.38 (1H, s, H-10), 6.31 (1H, brs, H-7), 3.89 (3H, s, 6-CH_3_O), 3.01 (1H, brs, H-3a), 2.76 (1H, brs, H-3b), 1.65 (3H, s, 2-CH_3_).

^13^C-NMR (150 MHz, CD_3_OD) *δ*: 198.8 (C-4), 165.6 (C-5), 163.0 (C-6), 162.1 (C-8), 155.0 (C-10a), 144.9 (C-9a), 107,2 (C-5a), 104,0 (C-4a), 102.8 (C-9), 102.8 (C-10), 100.9 (C-2), 97.5 (C-7), 56.2 (6-CH_3_O), 48.2 (C-3), 28.1 (2-CH_3_).

*Ustilaginoidin A* (**2**). ^1^H-NMR (600 MHz, CDCl3/CD_3_OD, 3:7) *δ*: 6.90 (2H, s, H-10, H-10′), 6.41 (2H, s, H-7, H-7′), 5.98 (2H, s, H-3, H-3′), 3.94 (6H, s, 6-CH_3_O, 6′-CH_3_O), 2.36 (6H, s, 2-CH_3_, 2′-CH_3_).

^13^C-NMR (150 MHz, CDCl3/CD_3_OD, 3:7) *δ*: 185.4 (C-4, C-4′), 169.4 (C-2, C-2′), 164.8 (C-5, C-5′), 161.8 (C-6, C-6′), 160.8 (C-8, C-8′), 153.9 (C-10a, C-10′a), 142.3 (C-9a, C-9′a), 108.2 (C-5a, C-5′a), 107.4 (C-3, C-3′), 104.3 (C-4a, C-4′a), 101.4 (C-10, C-10′), 101.9 (C-9, C-9′), 98.2 (C-7, C-7′), 56.4 (6-CH_3_O, 6′-CH_3_O), 20.7 (2-CH_3_, 2′-CH_3_).

*Fonsecin B* (**3**). ^1^H-NMR (600 MHz, CDCl3/CD_3_OD, 9:1) *δ*: 6.46 (1H, s, H-10), 6.42 (1H, d, *J* = 2.1 Hz, H-9), 6.23 (1H, d, *J* = 2.1 Hz, H-7), 3.88 (3H, s, 6-CH_3_O), 3.82 (3H, s, 8-CH_3_O), 2.93 (1H, d, *J* = 20.0 Hz, H-3a), 2.80 (1H, d, *J* = 20.0 Hz, H-3b), 1.63 (3H, s, 2-CH_3_).

^13^C-NMR (150 MHz, CDCl3/CD_3_OD, 9:1) *δ*: 197.1 (C-4), 164.2 (C-5), 162.4 (C-8), 161.3 (C-6), 153.8 (C-10a), 143.4 (C-9a), 107.2 (C-5a), 103.3 (C-4a), 102.7 (C-10), 99.9 (C-2), 98.6 (C-9), 96.7 (C-7), 56.0 (6-CH_3_O), 55.4 (8-CH_3_O), 47.5 (C-3), 27.8 (2-CH_3_).

*Dianhydroaurasperone C* (**4**). ^1^H-NMR (600 MHz, CDCl_3_) *δ*: 7.14 (1H, s, H-7′), 6.99 (1H, s, H-6′), 6.48 (1H, d, *J* = 2.2 Hz, H-9), 6.36 (1H, brs, H-2), 6.32 (1H, brs, H-3′), 6.25 (1H, d, *J* = 2.2 Hz, H-7), 4.02 (3H, s, 10-CH_3_O), 3.63 (3H, s, 8-CH_3_O), 3.61 (3H, s, 10′-CH_3_O), 2.56 (3H, brs, 2-CH_3_), 2.47 (3H, brs, 2′-CH_3_).

^13^C-NMR (150 MHz, CDCl_3_) *δ*: 167.1 (C-2), 166.8 (C-2′), 162.5 (C-8), 159.9 (C-10), 157.1 (C-10′), 156.5 (C-5′), 156.2 (C-8′), 116.1 (C-9′), 110.9 (C-3′), 110.7 (C-3), 109.5 (C-4′a), 108.4 (C-4a), 108.4 (C-10′a), 106.2 (C-7′), 105.8 (C10a), 97.6 (C-9), 96.4 (C-7), 61.8 (10′-CH_3_O), 56.3 (10-CH_3_O), 55.5 (8-CH_3_O), 20.8 (2′-CH_3_), 20.8 (2-CH_3_).

*Asperpyrone E* (**5**). ^1^H-NMR (600 MHz, CDCl_3_) *δ*: 7.14 (1H, s, H-7), 6.99 (1H, s, H-6′), 6.48, (1H, d, *J* = 2.2 Hz, H-9), 6.36 (1H, brs, H-3), 6.32 (1H, brs, H-3′), 6.25, (1H, d, *J* = 2.2 Hz, H-7), 4.02 (3H, s, 10-CH_3_O), 3.63 (3H, s, 8-CH_3_O), 3.61 (3H, s, 10′-CH_3_O), 2.56 (3H, brs, 2-CH_3_), 2.47 (3H, brs, 2′-CH3).

^13^C-NMR (150 MHz, CDCl_3_) *δ*: 167.1 (C-2), 166.8 (C-2′), 162.5 (C-8), 159.9 (C-10), 157.1 (C-10′), 156.5 (C-5′), 156.2 (C-8′), 116.1 (C9′), 110.9 (C-3′), 110.7 (C-3), 109.5 (C-4′a), 108.4 (C10′a), 108.4 (C-4a), 106.2 (C-7′), 105.9 (C-6′), 105.8 (C10a), 97.6 (C-9), 96.4 (C-7), 61.8 (10′-CH_3_O), 56.3 (10-CH_3_O), 55.5 (8-CH_3_O), 20.8 (2-CH_3_), 20.8 (2′-CH_3_).

*Fonsecinone A* (**6**). ^1^H-NMR (600 MHz, CD_3_OD) *δ*: 7.23 (1H, s, H-7), 7.12 (1H, s, H-6), 6.43, (1H, brs, H-3), 6.53 (1H, d, *J* = 2.0 Hz, H-7′), 6.25 (1H, d, *J* = 2.0 Hz, H-9′), 6.10 (1H, brs, H-3′), 3.99 (3H, s, 6′-CH_3_O), 3.81 (3H, s, 8-CH_3_O), 3.59 (3H, s, 8′-CH_3_O), 3.48 (3H, s, 10-CH_3_O), 2.53 (3H, brs, 2-CH_3_), 2.16 (3H, brs, 2′-CH3).

^13^C-NMR (150 MHz, CD_3_OD) *δ*: 182.3 (C-4′), 170.0 (C-2′), 169.6 (C-2), 163.1 (C-8′), 162.1 (C-6′), 161.2 (C-8), 157.6 (C-10), 157.2 (C-10b), 118.3 (C-9), 110.8 (C-3), 110.0 (C-4a), 108.9 (C10a), 108.7 (C5′a), 107.5 (C-3′), 106.9 (C-10′), 106.8 (C-6), 105.0 (C4′a), 102.8 (C-7), 98.0 (C-7′), 97.2 (C-9′), 61.7 (10-CH_3_O), 56.3 (8-CH_3_O), 56.3 (6′-CH_3_O), 55.4 (8′-CH_3_O), 20.1 (2-CH_3_), 20.1 (2′-CH_3_).

*Aurasperone A* (**7**). ^1^H-NMR (600 MHz, CD_3_OD) *δ*: 7.37 (1H, s, H-10), 7.26 (1H, s, H-3), 6.51 (1H, d, *J* = 2.1 Hz, H-7′), 6.25 (1H, d, *J* = 2.1 Hz, H-9′), 6.15 (1H, brs, H-2), 6.09 (1H, brs, H-3′), 3.98 (3H, s, 6′-CH_3_O), 3.81 (3H, s, 8-CH_3_O), 3.61 (3H, s, 8′-CH_3_O), 3.48 (3H, s, 6-CH_3_O), 2.45 (3H, brs, 2-CH_3_), 2.16 (3H, brs, 2′-CH_3_).

^13^C-NMR (150 MHz, CD_3_OD) *δ*: 185.8 (C-4), 170.4 (C-2), 169.9 (2′), 162.9 (8′), 161.9 (6′), 161.4 (8), 158.9 (6), 154.3 (10a), 142.3 (9a), 118.5 (7), 111.7 (**5a**), 108.8 (5′a), 107.7 (C-3), 107.4 (3′), 106.7 (10′), 105.2 (C**4a**), 104.5 (4′a), 102.8 (9), 102.8 (10), 98.0 (7′), 97.2 (9′), 62.3 (6-CH_3_O), 56.3 (8-CH_3_O), 56.3 (6′-CH_3_O), 55.4 (8′-CH_3_O), 20.3 (2-CH_3_), 20.1 (2′-CH_3_).

### 2.7. Radical Scavenging Capacity Assay

A colorimetric assay was adapted from Re et al. [15]. It consists of the measurement of the decolorization of radical monocation ABTS^•+^ as it reacts with the NγPs. The decolorization was monitored in a 96-well microplate using a microplate reader Infinite 200 (Tecan©, Mennedorf, Switzerland). The reaction between the sample and the ABTS^•+^ solution was performed for 5 min at 25 °C and the reading was made at 734 nm. The scavenging activity is expressed in Trolox equivalent antioxidant capacity (TEAC). A calibration curve of 3,4-dihydro-6-112 hydroxy-2,5,7,8-tétramethyl-2H-1-benzopyran-2-carboxylic acid (Trolox) ranging from 15.625 to 1000 μmol L^−1^ was made. To compare the various TEAC, the concentrations of the sample were set to 10 μmol L^−1^.
(1)$$\mathrm{TEAC}\;=\;\frac{{\left(A_{0}\;-{\;A}_{f}\right)}_{\mathrm{sample}}\;-\;{\left(A_{0}\;-{\;A}_{f}\right)}_{\mathrm{blank}}}{{\left(A_{0}\;-{\;A}_{f}\right)}_{\mathrm{Trolox}}\;-\;{\left(A_{0}\;-{\;A}_{f}\right)}_{\mathrm{blank}}}$$
where A_0_ is the initial absorbance and A_f_ the absorbance after 5 min.

### 2.8. Cell Innocuity and Protective Effect Against H_2_O_2_-Mediated Cell Death

Innocuity was evaluated on normal/non-cancerous cells, i.e., Chinese hamster ovary cells (CHO, ATCC©, Manassas, VA, USA). Cells were routinely grown on 25 cm^2^ flasks in Dulbecco’s modified essential medium (DMEM) supplemented with 10% fetal calf serum (FCS), 1% l-glutamine, and 1% antibiotics (Invitrogen©, Carlsbad, CA, USA), and maintained in a 5% CO_2_ incubator at 37 °C. For innocuity testing, CHO cells grown on flasks were trypsinized, diluted in culture media, and seeded onto 96-well plates (Greiner Bio-One©, Austria) at 20,000 cells per well [16]. After seeding, cells were left to reach confluence for 48 h. Confluent cells were then exposed to increasing doses of compounds **1**–**7** or ascorbic acid for 24 h at 37 °C before evaluation of the cell viability using a resazurin-based in vitro toxicity assay kit (Sigma-Aldrich), as previously described [17,18].

Briefly, resazurin stock solution was diluted 1:100 in sterile PBS containing calcium and magnesium (PBS++, pH 7.4). Plates were aspirated and 100 μL of the diluted solution was added per well. After 1 h incubation at 37 °C, fluorescence intensity was measured using a microplate reader (Biotek, Winnooski, VT, USA, Synergy Mx) with an excitation wavelength of 530 nm and an emission wavelength of 590 nm. The fluorescence values were normalized by the controls (untreated cells) and expressed as percent viability.
(2)$$\mathrm{Cell}\;\mathrm{viability}\;\left(\%\;\mathrm{of}\;\mathrm{control}\right)\;=\;\frac{A_{\mathrm{sample}}\;-{\;A}_{\mathrm{blank}}}{A_{\mathrm{control}}\;-{\;A}_{\mathrm{blank}}}\;\times \;100\;\;$$

For protective effect against H_2_O_2_-mediated cell death, confluent cells on 96-well plates were exposed for 30 min to increasing doses of compounds **1**–**7** or ascorbic acid before treatment with 300 μmol L^−1^ of hydrogen peroxide for 24 h at 37 °C prior to evaluation of the cell viability using resazurin-based assay, as explained above. Results were expressed as percentage of inhibition of the hydrogen peroxide effect on the cells.
(3)$$\%\;\mathrm{of}\;\mathrm{inhibition}\;\mathrm{of}\;\mathrm{the}\;H_{2}O_{2}\;\mathrm{effect}\;\;=\;100-\;\frac{A_{\mathrm{sample}}\;-{\;A}_{\mathrm{blank}}}{A_{\mathrm{control}}\;-{\;A}_{\mathrm{blank}}}\;\times \;100\;\;$$

### 2.9. Data Analysis

All experiments were performed in triplicate. The results were fitted using R software version 3.5.1.

## 3. Results

### 3.1. Isolation of the NγPs

Most of the studies evaluating production of NγPs involve a microorganism growing on synthetic media [2,11,13,19,20,21]. In the present case, the culture was performed on an optimized solid medium containing solid agroindustrial byproducts, since these culture conditions were found to strongly increase NγP production, allowing the isolation of all produced NγPs, including the less abundant ones [22]. From the fungal ethanolic extract, seven NγPs were isolated by semi-preparative HPLC after liquid–liquid extraction (Figure 1), including: Fonsecin (**1**), ustilaginoidin A (**2**), fonsecin B (**3**), dianhydroaurasperone C (**4**), asperpyrone E (**5**), fonsecinone A (**6**), and aurasperone A (**7**) [20,23,24,25,26,27]. The structures and carbon numbering used for NMR assignment are given in Figure 2. Compounds **1** and **3** are monomeric NγPs and compounds **4**–**7** are dimeric NγPs. The NγPs contain a fully conjugated system, giving rise to very characteristic UV/VIS spectra, with an absorption maximum at 280 nm.

### 3.2. ABTS Assay

Table 1 gives the AOA reported for NγPs in the literature. The radical scavenging activity against ABTS^•+^ was tested for the seven NγPs isolated at a concentration of 10 μmol L^−1^. Compounds **1**–**7** exhibited various AOA activity, expressed in TEAC, i.e., 13.3, 14.6, 5.1, 4.6, 3.3, 2.4, to 2.4 μmol L^−1^ respectively (Table 1). In comparison, ascorbic acid showed a TEAC of 13.2 μmol L^−1^, very close to the ones observed for foncesin (**1**) and ustilaginoidin A (**2**).

### 3.3. Structure–Mechanism Relationship

A linear relation between the degree of substitution with hydroxyl groups per NγP unit and the TEAC was observed for the isolated NγPs (Figure 3). The degree of substitution with hydroxyl groups per NγP unit is presented in Table 1.

The proposed radical scavenging mechanism is based on a resonance stabilization of the unpaired electron through its redistribution on the aromatic cores of the NγPs (Figure 4).

### 3.4. CHO Cell-Based Assay

The innocuity and the protective effect of the various compounds against H_2_O_2_-mediated cell death were investigated using an in vitro model, i.e., CHO cells. With regards to the innocuity of the compounds, only compounds **1** and **2**—with the highest radical scavenging potential—reduced the cells’ viability at higher doses with 56.5 and 35% reduction in cell viability for 3.4 and 1.9 mmol L^−1^ of compounds (Figure 5). Compounds **3**–**7** did not influence the cell viability, even at the highest doses tested (ranging from 1 to 3 mmol L^−1^).

CHO cells were exposed for 24 h to increasing concentrations of compounds **1**–**7** or ascorbic acid. At the end of the incubation period, cell viability was evaluated using Alamar blue assay. Error bars represent the variability associated with the measurements of triplicate samples.

About the protective effect of compounds **1**–**7** against H_2_O_2_-mediated cell death, and as expected, incubation of CHO cells for 24 h with 300 μmol L^−1^ of H_2_O_2_ decreases cell viability by 75.5 ± 0.7%. Pre-exposure of CHO cells to increasing concentrations of compounds **1**–**7** or ascorbic acid resulted in variable degrees of protection against H_2_O_2_ (Figure 6).

CHO cells were exposed for 30 min to increasing concentrations of compounds **1**–**7** or ascorbic acid in the presence of 300 μmol L^−1^ of H_2_O_2_. At the end of the incubation period, cell viability was evaluated using Alamar blue assay. Error bars represent the variability associated with the measurements of triplicate samples.

Results show that the NγPs with a higher degree of substitution with hydroxyl groups exhibited a better protective effect. Indeed, a strong protective effect was measured for compounds **1**, **2**, and **3** with IC_50_ on H_2_O_2_-mediated cell death of 3.7, 2.25, and 189.0 μmol L^−1^, respectively. Compounds **4**, **6**, **7**, and ascorbic acid exhibited a lower protective effect with IC_50_ on H_2_O_2_-mediated cell death of 1.79 mmol L^−1^ for compound **4**, and IC_50_ > 1.8 mmol L^−1^ for compounds **5**, **6**, and **7**.

## 4. Discussion

Compounds **1** and **3**–**7** have already been characterized in *Aspergillus* genus [20,24,28,29]. Interestingly, compound **2**, which is a chaetochromin-type NγP, has been reported from the genus of *Chaetomium*, *Fusarium*, *Penicillium*, *Metarhizium*, and *Ustilaginoidea*, but is very scarce in *Aspergillus* spp., which is characteristic of asperpyrone-type NγPs and nigerone-type NγPs [3,30,31,32]. Frisvad et al. made the hypothesis that an *Aspergillus flavus* strain producing ustilaginoidin C could be the consequence of horizontal transfer of gene clusters coding for ustilaginoidins from one fungus to another, while occupying the same ecological niche during evolution [33].

Concerning the AOA, fonsecin has been found to exhibit potent radical scavenging activity against DPPH radical, which is coherent with the results of Leutou et al. [11]. Zhang et al. found no significant AOA for aurasperone A, dianhydroaurasperone C, or fonsecinone A, whereas we report AOA for these compounds [13]. An explanation of this difference could be that these authors used concentrations of 0.088, 0.090, and 0.088 μmol L^−1^ respectively, which are 100 times less concentrated that the one we used for these compounds. Accordingly, Cai et al. also reported AOA for aurasperone A and fonsecinone A using a similar protocol to the one used by Zhang et al., but using concentrations 5000 time higher than them [10,13]. It is, however, difficult to compare results obtained with DPPH and ABTS protocols, although both colorimetric methods use the same single electron transfer mechanism [34,35].

Although the relationship between the structure of NγPs and their biological activities is poorly described in the literature, hypotheses have been made concerning the mechanism behind this activity, while observing the structure of other antioxidant such as naphtho-α-pyrenes or flavonoids [6]. These studies suggest that the antioxidant capacity of such molecules is due to the presence of an aromatic core—undergoing a resonance structure, able to stabilize the unpaired electron through its redistribution on the aromatic core—and number of free hydroxyl groups, able to donate a hydrogen radical and thereby scavenge the free radicals. For that matter, the antioxidant capacity is a function of the number and the position of the hydroxyl groups on the molecule [36,37]. This hypothesis is verified for the NγPs isolated here, as there is a linear relation between the degree of substitution with hydroxyl groups per NγP unit and the radical scavenging activity. In comparison with the results found in the literature, a graphic representation of the degree of substitution with hydroxyl groups per NγP unit and the logarithmic values of the IC_50_ obtained by Leutou et al. shows that, except for aurasperone A, there is a linear relation between these variables [11]. No linear relation appears in the case of Cai et al., but the range of degree of substitution with hydroxyl groups per NγP unit they investigated was smaller (varying between 1 and 1.5) [10]. However, the variability of values they obtained suggests that for the same number of hydroxyl groups on the molecule, the position of these groups also influences the scavenging activity. No such linear relation could have been highlighted for the results of Zhang et al., although the two highest AOA values they obtained correspond to the NγPs with the highest degree of substitution with hydroxyl groups [13].

With regards to the innocuity of the compounds, only the compounds with the highest radical scavenging potential reduce the cell viability at higher doses. Accordingly, Zhan et al. tested various NγPs—including the present fonsecin, fonsecin B, fonsecinone A, aurasperone A, and dianhydroaurasperone C—and none of them showed cytotoxicity against the human cancer cell lines NCI-H460 (non-small cell lung carcinoma), MIA Pa Ca-2 (pancreatic cancer), MCF-7 (breast cancer), and SF-268 (CNS cancer; glioma) [2]. Following the same idea, Zhang et al. showed that none of their tested NγPs—including the present aurasperone A, dianhydroaurasperone C, and fonsecinones A—exhibited cytotoxic effect on the tumor cell lines SMMC-7721 (hepatocellular carcinoma cells) and A549 (human lung epithelial cells) at a dose ranging from 32.97 to 66.23 μmol.L^−1^ [13]. In accordance with our results concerning the protective effect of the isolated compounds on CHO against H_2_O_2_, NγP glycosides from *Cassia obtusifolia* have been shown to protect human liver-derived HepG2 cells against the toxicity of tert-butylhydroperoxide—an oxidative stress inducer—by significantly inhibiting the reactive oxygen species generation [38].

The evaluation of the radical scavenging potential of a given substance on cellular models might be a good requisite for eventual applications in cosmetics [39,40]. It is also good complementary information in addition to the ABTS-based antioxidant measurement. Indeed, it helps to underline mechanisms in complex systems; to be effective, the antioxidants need to be absorbed, transported, distributed, and retained properly in the biological fluids, cells, and tissues [41]. In addition, to be used in food, medicine, or cosmetics, the antioxidant molecule must be safe/non-toxic. The innocuity and the protective effect of the various compounds against H_2_O_2_-mediated cell death were investigated using an in vitro model, i.e., CHO cells. Indeed, since the Free Radical Theory of Aging, attention has been focused on the role of reactive oxygen species (ROS) in the initiation and progression of the aging process [42]. ROS are known to cause damage to molecules such as DNA, proteins, and lipids, and for that reason, ROS-induced damage contributes to the natural aging process—as ROS are naturally produced in the cells—but also in a variety of biological phenomena, including radiation damage, infection with xenobiotics, carcinogenesis, and neurodegenerative diseases among others [43]. Among the ROS of biological interest, the hydrogen peroxide can undergo Fenton-type reactions with metals such as Fe^2+^ to produce OH^•^ radicals. These radicals are the most noxious and interactive of the radical species and are known to initiate rounds of peroxidative damage to membrane lipids [44]. Hence, present in critical concentration, H_2_O_2_ in the cytosol induces an oxidative stress that eventually results in cell death [45]. Protective effects of a compound against H_2_O_2_-mediated cell death in normal/non-cancerous cells such as CHO cells is a good way to assess the efficiency of antioxidant potential of this compound in biological conditions [46].

Finally, although some studies have been carried out on this strain to understand its metabolism in order to optimize NγP production, further analyses are required in the case of an industrial application [22,47].

## 5. Conclusions

NγPs are fungal secondary metabolites exhibiting interesting radical scavenging properties. This effect has been documented in vitro for some members of this family. In this study, we demonstrated that seven NγPs isolated from *A. tubingensis* G131 have different antioxidant effect using ABTS and cell-based assays. Amongst all of the antioxidant NγPs, fonsecin and ustilaginoidin A exhibit interesting results, particularly against ROS-mediated cell death. The hypothesis behind this increased radical scavenging potential is that the higher the number of hydroxyl substitutes on the NγP skeleton is, the higher this potential will be. Such properties could be applied in the food and cosmetic industries.

## Figures and Tables

**Figure 1 biomolecules-10-00029-f001:**
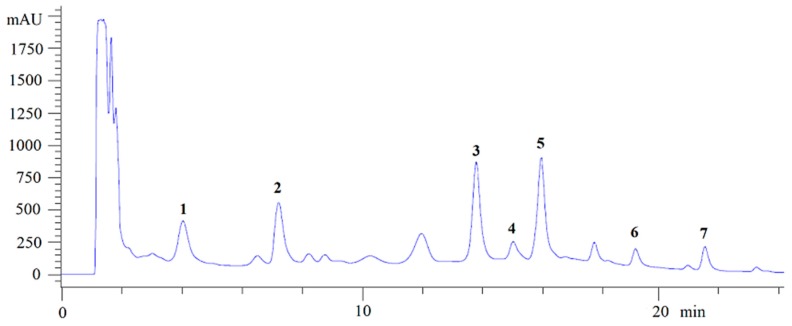
Chromatogram of the ethanolic extract with molecules associated to the peaks.

**Figure 2 biomolecules-10-00029-f002:**
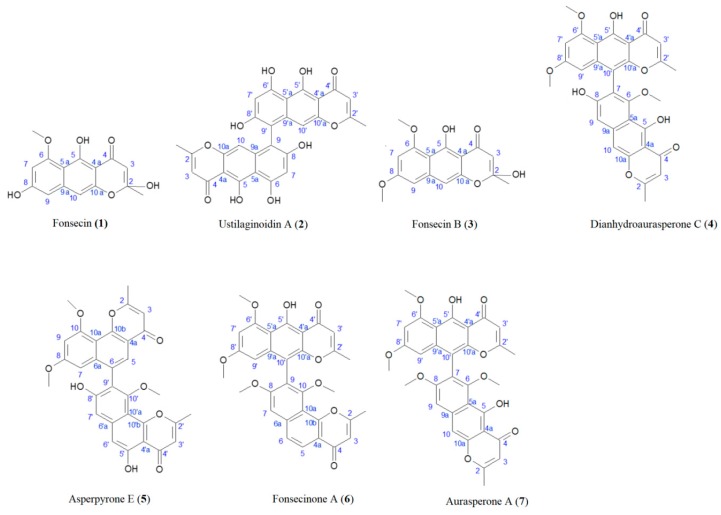
Structures of compounds **1**–**7** and carbon numbering used for NMR assignment.

**Figure 3 biomolecules-10-00029-f003:**
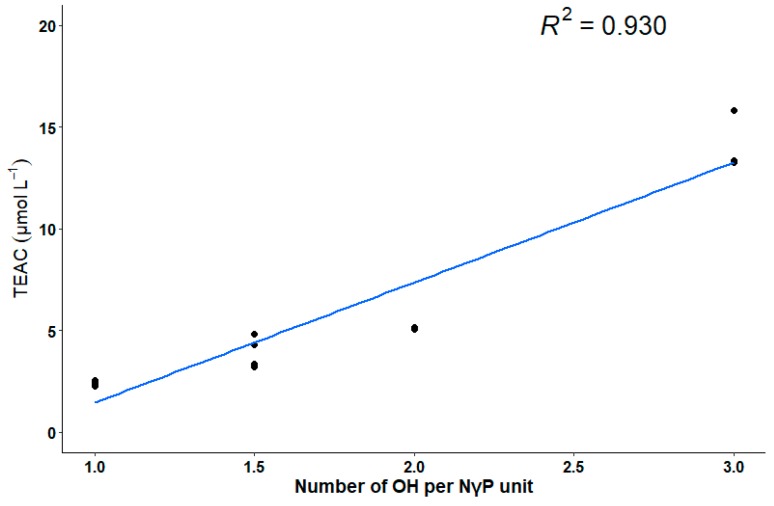
Linear relation between the number of hydroxyl groups per NγP unit and the TEAC.

**Figure 4 biomolecules-10-00029-f004:**
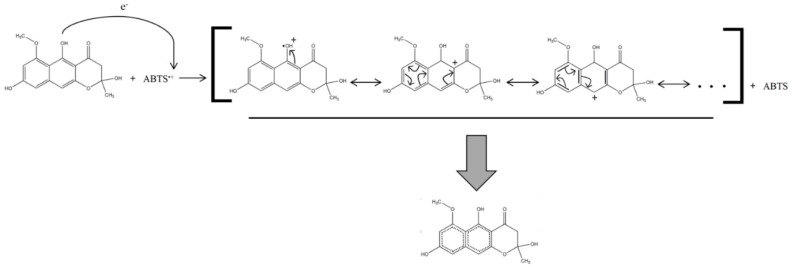
Radical scavenging mechanism of the fonsecin.

**Figure 5 biomolecules-10-00029-f005:**
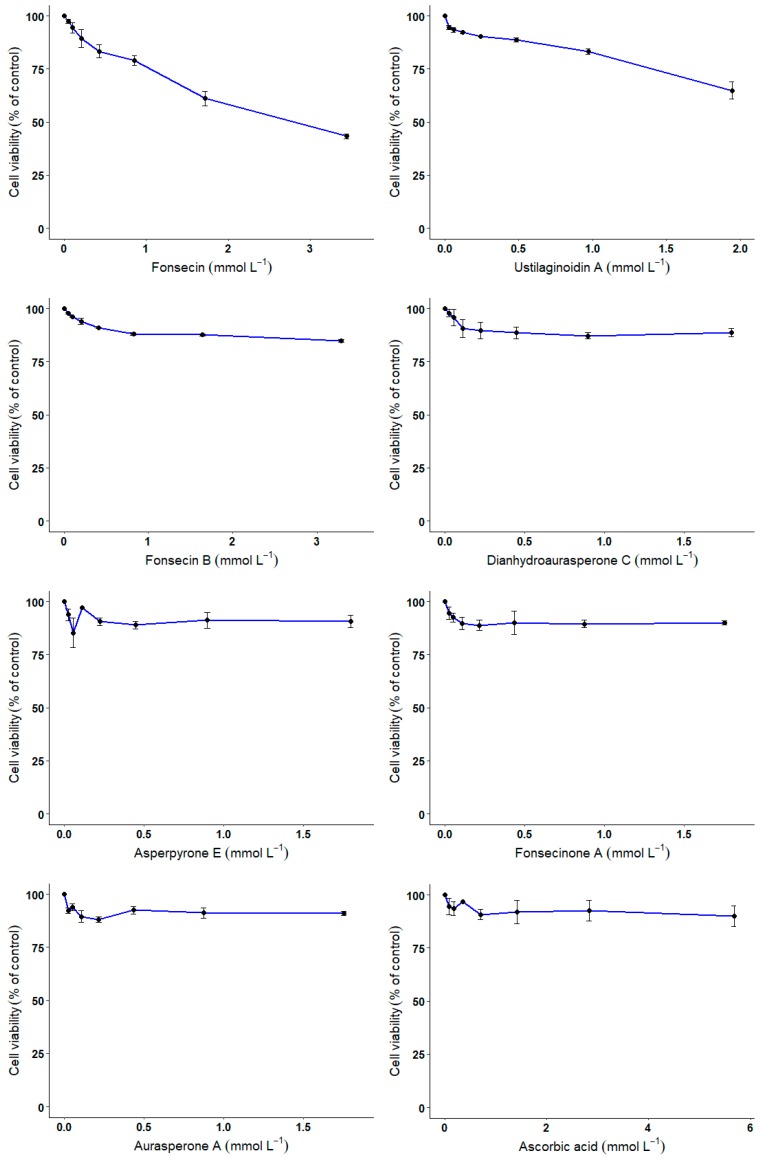
Innocuity testing of compounds **1**–**7** on Chinese hamster ovary (CHO) cells.

**Figure 6 biomolecules-10-00029-f006:**
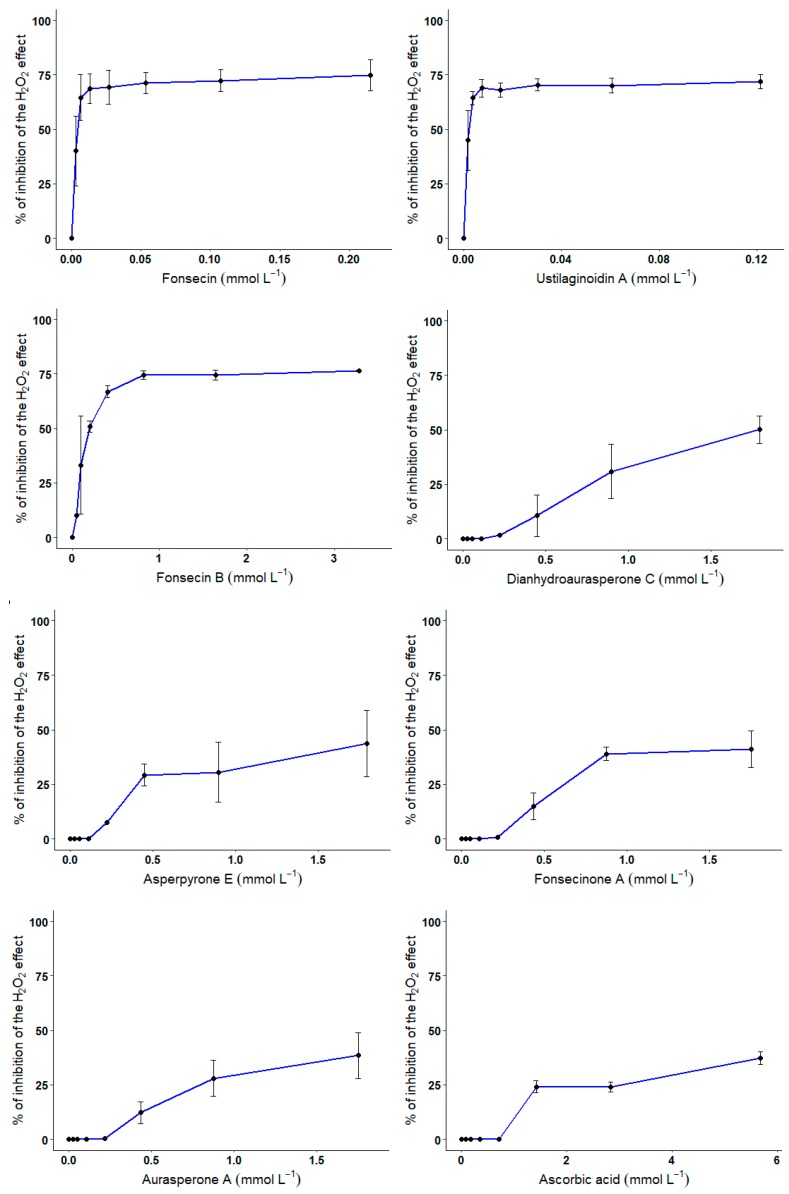
Protective effect of compounds **1**–**7** on H_2_O_2_-mediated cell death in CHO cells.

**Table 1 biomolecules-10-00029-t001:** Antioxidant activity (AOA) of the NγPs reported in the literature.

Compounds	Number of OH Per NγP Unit	ABTS	AOA DPPH Essay			
		TEAC (µmol L^−1^), Tested Concentration: 10 µmol L^−1^ (in This Work)	IC_50_ (µmol L^−1^) [11]	Radical Scavenging (%), Tested Concentration: 50 µg L^−1^ [13]	[12]	Radical Scavenging (%), Tested Concentration: 250,000 µg L^−1^ [10]
6,9-Dibromoflavasperone	1		21.0			
Ascorbic acid		13.23 ± 0.05 ^a^	20.0			90.5
Asperpyrone A	1			N		
Asperpyrones B	1					31.9
Asperpyrone C	1			N		32.9
Asperpyrone E	1	3.29 ± 0.11 ^c^				
Asperpyrone F	1.5					32.8
Aurasperone A	1	2.4 ± 0.20 ^d^		N	N	38.6
Aurasperone B	2		0.01	48.1		
Aurasperone E	1					33.4
BHT				80.4		
Dianhydroaurasperone C	1.5	4.57 ± 0.36 ^b,c^		N		
Flavasperone	1		25.0		N	
Fonsecin	3	13.32 ± 0.08 ^a^	0.02			
Fonsecin B	2	5.12 ± 0.08 ^b^				
Fonsecinone A	1	2.36 ± 0.07 ^d^		N		36.9
Fonsecinones B	1.5			13.7		38.7
Fonsecinone C	1.5			N		
Fonsecinone D	1.5			37.5	N	
Nigerone A	2			N		
Nigerone B	2			N		
Nigerone C	2			41.6		
Rubrofusarin B	1				N	
TMC-256A1	2		0.30			
Ustilaginoidin A	3	14.59 ± 1.75 ^a^				

TEAC: Trolox equivalent antioxidant capacity; N: No significant activity; different letters in superscript ^a,b,c,d^ indicate a significant difference, measured with a *t*-test at a risk α = 0.05.

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
