# Peer review of "Naphtho-Gamma-Pyrones Produced by *Aspergillus tubingensis* G131: New Source of Natural Nontoxic Antioxidants"

_biomolecules, 2019, doi:10.3390/biom10010029_

Round 1
Reviewer 1 Report
It is opinion of the reviewer that this paper before acceptance needs minor revision. My individual comments are listed below.
20 – Remove „assay measured”.
22, 24 – It should be „H2O2”.
24 – It should be “IC50”.
53 – It should be “…using the 2,2’….. –sulfonic acid radical cation (ABTS●+) assay …”.
75 – Supplier of fungal material?
94 – “Compounds 1101”?
101 – It should be “The reaction between NγPs and the ABTS●+ “?
111 – Reference for 2.8 is needed.
Figure 3 – R2 should be reported with 3 digitals after decimal point.
202 – It should be “against DPPH radical”.
355 – “: Antioxidant activity ….” Must be removed”.
Elucidation of the chemical structure of compounds 1-7 must described (Results of NMR).
Author Response
Reviewer 1
First, we would like to thank the reviewer for his/her comments
20 – Remove „assay measured”.
According to reviewer’s suggestion, we removed it.
22, 24 – It should be „H2O2”.
According to reviewer’s suggestion we changed it.
24 – It should be “IC50”.
According to reviewer’s suggestion we changed it.
53 – It should be “…using the 2,2’….. –sulfonic acid radical cation (ABTS●+) assay …”.
According to reviewer’s suggestion we changed it.
75 – Supplier of fungal material?
In the revised version we precised the supplier “was provided by École Nationale Supérieure Agronomique de Toulouse (France) ”
94 – “Compounds 1101”?
According to reviewer’s suggestion we changed for “compounds 1”.
101 – It should be “The reaction between NγPs and the ABTS●+ “?
According to reviewer’s suggestion we changed it as suggested.
111 – Reference for 2.8 is needed.
In the revised version, we added a reference.
Figure 3 – R2 should be reported with 3 digitals after decimal point.
According to reviewer’s suggestion we added another digit.
202 – It should be “against DPPH radical”.
According to reviewer’s suggestion we changed it
355 – “: Antioxidant activity ….” Must be removed”.
According to reviewer’s suggestion we removed it.
Elucidation of the chemical structure of compounds 1-7 must described (Results of NMR)
In the revised manuscript we added some information in the material & methods part, added the spectroscopic data of the NMR and the references.
Regards

Reviewer 2 Report
The Manuscript reports new and relevant information about naphtho-gamma-pyrones from Aspergillus tubingensis. The Manuscript is acceptable for publication with minor modifications, as detailed below.
TABLE 1
-The experimental data must be reported as means +- standard deviations of the triplicate assays declared by the authors in Materials and Methods section (2.9 Data Analysis). Additional letters close to numeric results must be included. Distinct letters will indicate different values with statistical significance (p<0.05). These letters can be obtained from ANOVA with the similarity test of Duncan/Dunnet/Tukey, using the SPSS or equivalent software.
REFERENCES
Some recent and relevant References must be included and commented in the Introduction and/or the Results and/or Discussion sections.
-Obermaier, S., and Muller, M. (2019). Biaryl-Forming Enzymes from Aspergilli Exhibit Substrate Dependent Stereoselectivity. Biochemistry 58, 2589-2593.
-Lima, M.A.S., de Oliveira, M.D.F., Pimenta, A.T.A., and Uchoa, P.K.S. (2019). Aspergillus niger: A Hundred Years of Contribution to the Natural Products Chemistry. J Braz Chem Soc 30, 2029-2059.
-Lilia, L.D., Isaura, C., Julie, B., Elodie, C., Jose, R., Patricia, T., and Florence, M. (2019). Influence of Culture Conditions on Production of NGPs by Aspergillus tubingensis. J Microbiol Biotechnol 29, 1412-1423.
-Choque, E., Klopp, C., Valiere, S., Raynal, J., and Mathieu, F. (2018). Whole-genome sequencing of Aspergillus tubingensis G131 and overview of its secondary metabolism potential. BMC Genomics 19, 16.
-Ma, Y.M., Li, T., and Ma, C.C. (2016). A new pyrone derivative from an endophytic Aspergillus tubingensis of Lycium ruthenicum. Nat Prod Res 30, 1499-1503.
-Huang, H.B., Feng, X.J., Liu, L., Chen, B., Lu, Y.J., Ma, L., She, Z.G., and Lin, Y.C. (2010). Three Dimeric Naphtho-gamma-Pyrones from the Mangrove Endophytic Fungus Aspergillus tubingensis Isolated from Pongamia pinnata. Planta Med 76, 1888-1891.
Best regards.
Author Response
Reviewer 2
The Manuscript reports new and relevant information about naphtho-gamma-pyrones from Aspergillus tubingensis. The Manuscript is acceptable for publication with minor modifications, as detailed below.
We thank the reviewer for his/her comments.
TABLE 1
-The experimental data must be reported as means +- standard deviations of the triplicate assays declared by the authors in Materials and Methods section (2.9 Data Analysis). Additional letters close to numeric results must be included. Distinct letters will indicate different values with statistical significance (p<0.05). These letters can be obtained from ANOVA with the similarity test of Duncan/Dunnet/Tukey, using the SPSS or equivalent software.
According to reviewer’s suggestion we changed the table following the recommendations.
REFERENCES
Some recent and relevant References must be included and commented in the Introduction and/or the Results and/or Discussion sections.
-Obermaier, S., and Muller, M. (2019). Biaryl-Forming Enzymes from Aspergilli Exhibit Substrate Dependent Stereoselectivity. Biochemistry 58, 2589-2593.
-Lima, M.A.S., de Oliveira, M.D.F., Pimenta, A.T.A., and Uchoa, P.K.S. (2019). Aspergillus niger: A Hundred Years of Contribution to the Natural Products Chemistry. J Braz Chem Soc 30, 2029-2059.
-Lilia, L.D., Isaura, C., Julie, B., Elodie, C., Jose, R., Patricia, T., and Florence, M. (2019). Influence of Culture Conditions on Production of NGPs by Aspergillus tubingensis. J Microbiol Biotechnol 29, 1412-1423.
-Choque, E., Klopp, C., Valiere, S., Raynal, J., and Mathieu, F. (2018). Whole-genome sequencing of Aspergillus tubingensis G131 and overview of its secondary metabolism potential. BMC Genomics 19, 16.
-Ma, Y.M., Li, T., and Ma, C.C. (2016). A new pyrone derivative from an endophytic Aspergillus tubingensis of Lycium ruthenicum. Nat Prod Res 30, 1499-1503.
-Huang, H.B., Feng, X.J., Liu, L., Chen, B., Lu, Y.J., Ma, L., She, Z.G., and Lin, Y.C. (2010). Three Dimeric Naphtho-gamma-Pyrones from the Mangrove Endophytic Fungus Aspergillus tubingensis Isolated from Pongamia pinnata. Planta Med 76, 1888-1891.
According to reviewer’s suggestion we added the following references :
Obermaier, S., and Muller, M. (2019). Biaryl-Forming Enzymes from Aspergilli Exhibit Substrate Dependent Stereoselectivity. Biochemistry 58, 2589-2593.
-Lilia, L.D., Isaura, C., Julie, B., Elodie, C., Jose, R., Patricia, T., and Florence, M. (2019). Influence of Culture Conditions on Production of NGPs by Aspergillus tubingensis. J Microbiol Biotechnol 29, 1412-1423.
-Choque, E., Klopp, C., Valiere, S., Raynal, J., and Mathieu, F. (2018). Whole-genome sequencing of Aspergillus tubingensis G131 and overview of its secondary metabolism potential. BMC Genomics 19, 16.
-Ma, Y.M., Li, T., and Ma, C.C. (2016). A new pyrone derivative from an endophytic Aspergillus tubingensis of Lycium ruthenicum. Nat Prod Res 30, 1499-1503.
Best regards.

Reviewer 3 Report
Please see attached

Author Response
First, we would like to thank the reviewer for his/her comments.
This is an interesting article that uses several instrumental means to support its experimental findings.
I am very unhappy with the fact that not even one chemical equation depicting the antioxidants’ transformation is presented. In order for this worthwhile study to be accepted by a larger audience such a request is imperative.
According to reviewer’s suggestion we added a figure (Figure 4) to explain the mechanism.
In addition a more extensive description of the NMR study is absolutely necessary (solvent?)
In the revised version, we added some information in the material & methods part, added the spectroscopic data of the NMR and the references.
Line 87 does not describe the way extraction was performed (Time? Stirring speed? Temperature?)
According to reviewer’s suggestion we precised the extraction conditions.
Some edits/typo corrections:
Lines 22 and 24: change the subscripts for hydrogen peroxide
According to reviewer’s suggestion we changed it.
Lines 21 and 25: superscript mL-1
Change was done in the revised version.
Line 23: change “for cells” to “towards cells”
According to reviewer’s suggestion we changed it.
Line 100: Change “oxydized in” to “oxidized to”
Change was done in the revised version.
Line 111: delete the second “on”
We removed it in the revised version.
Line 111: change “not-cancerous” to “non-cancerous”
According to reviewer comment, we changed it.
Line 120: change “μl” to “μL”
Change was done in the revised version.
Line 133: conducting the experiment in triplicate is not adequate as seen in the low R2 (0.93, line 164) value
We precised the error in the table in the revised version.
Line 168 and 230; change “regarding” with “with regards”
Change was done in the revised version.
Line 170: change “cells” to “ cells’ “ (add apostrophe)
Change was done in the revised version.
Line 198: change “do hypothesis” with “make the hypothesis”
Change was done in the revised version.
Line 215: provide a resonance stabilization series of structures to prove the point
This is now indicated in Figure 4 of the revised manuscript
Line 217: Change “is function” with “is a a function”
Change was done in the revised version.
Line 217: Change “atom” with ”radical”
Change was done in the revised version.
Line 270: Change “appreciated” with “applied”
Change was done in the revised version.
